# Evaluation of Vector Competence of *Ixodes* Ticks for Kemerovo Virus

**DOI:** 10.3390/v14051102

**Published:** 2022-05-20

**Authors:** Camille Victoire Migné, Hélène Braga de Seixas, Aurélie Heckmann, Clémence Galon, Fauziah Mohd Jaafar, Baptiste Monsion, Houssam Attoui, Sara Moutailler

**Affiliations:** 1ANSES, INRAE, Ecole Nationale Vétérinaire d’Alfort, UMR BIPAR, Laboratoire de Santé Animale, F-94700 Maisons-Alfort, France; camille.migne@anses.fr (C.V.M.); helene.bragadeseixas@outlook.fr (H.B.d.S.); aurelie.heckmann@anses.fr (A.H.); clemence.galon@anses.fr (C.G.); 2ANSES, INRAE, Ecole Nationale Vétérinaire d’Alfort, UMR 1161 Virologie, Laboratoire de Santé Animale, F-94700 Maisons-Alfort, France; fauziah.mohd-jaafar@vet-alfort.fr (F.M.J.); baptiste.monsion@vet-alfort.fr (B.M.)

**Keywords:** ticks, Kemerovo virus, tick-borne orbivirus, vector competence

## Abstract

Tick-borne viruses are responsible for various symptoms in humans and animals, ranging from simple fever to neurological disorders or haemorrhagic fevers. The Kemerovo virus (KEMV) is a tick-borne orbivirus, and it has been suspected to be responsible for human encephalitis cases in Russia and central Europe. It has been isolated from *Ixodes persulcatus* and *Ixodes ricinus* ticks. In a previous study, we assessed the vector competence of *I. ricinus* larvae from Slovakia for KEMV, using an artificial feeding system. In the current study, we used the same system to infect different tick population/species, including *I. ricinus* larvae from France and nymphs from Slovakia, and *I. persulcatus* larvae from Russia. We successfully confirmed the first two criteria of vector competence, namely, virus acquisition and trans-stadial transmission, for both tick species that we tested. The estimated infection rates of engorged and moulted ticks suggest specificities between viral strains and tick species/developmental stages.

## 1. Introduction

Ticks are the primary vectors of pathogens for both human and animal health in Europe. They can transmit the widest variety of microorganisms, including bacteria, parasites and viruses. Among these pathogens, 170 tick-borne viruses (TBVs) have been identified to date. They are classified into 9 virus families and 12 genera. TBV symptoms in humans and animals range from simple fever to neurological disorders or haemorrhagic fevers [1]. *Ixodes ricinus* is the most common tick in Europe and can transmit several pathogens that are responsible for severe diseases in human and animals, such as Lyme borreliosis, piroplasmosis and tick-borne encephalitis [2].

TBVs are arboviruses (arthropod-borne viruses), and their geographical distribution is correlated with their competent arthropod vectors [3]. KEMV is a tick-borne virus belonging to the genus *Orbivirus* within the family *Reoviridae.* It was discovered in 1962 and was first isolated from *I. persulcatus* ticks in the Kemerovo region of Russia [4]. It was also isolated from *I. ricinus* in 1964 in Czechoslovakia [5] and in 1975 in the Vologda region of Russia [4]. KEMV is suspected to be responsible for human encephalitis cases in Russia and central Europe [6,7]. Studying the vector competence of a tick species for a specific pathogen involves the validation of three criteria, namely, pathogen acquisition (during the first blood meal on an infected host), trans-stadial transmission (from one stage to the next one) and transmission to a susceptible naïve vertebrate host (upon a new blood meal on a host). In a previous study, we assessed the vector competence of *I. ricinus* ticks from Slovakia for KEMV. We used an artificial feeding system to infect larvae and validated the first two criteria of vector competence (virus acquisition and trans-stadial transmission) for this *I. ricinus* tick population. The last criterion was not confirmed [8]. In the current study, we used the same system to infect different tick populations from two distinct *Ixodes* tick species: *I. ricinus* larvae from France and nymphs from Slovakia, and *I. persulcatus* larvae from Russia.

## 2. Materials and Methods

### 2.1. Virus

BSR cells (a clone of BHK-21, [9]) were used to propagate the KEMV virus and to constitute a virus stock as previously described [8]. Passage 3 in BSR cells was designated KEMV smb1/Vero2/BSR3, and passage 4 was designated KEMV smb1/Vero2/BSR4 (smb—suckling mouse brain).

#### Plaque Assay

A KEMV titre was determined using a plaque assay as previously described [8,10]. Briefly, virus-infected cells (grown in 75 cm^2^ flasks) were harvested on day 5 post-infection, and cell debris was treated with an organic solvent to release virus particles as previously described [11]. The serially diluted virus was then titrated using BSR cells grown in 24-well plates. Plaques were stained with 0.1% naphthalene-black solution. Virus titres were calculated as 4.4 × 10^6^ PFU/mL for KEMV smb1/Vero2/BSR3 and 10^9^ PFU/mL for KEMV smb1/Vero2/BSR4.

### 2.2. Ticks

In this study, we used two different tick species: *I. ricinus* and *I. persulcatus* (Table 1). Prior to being used in the experimental procedure, a sample from each tick population was tested for the KEMV genome using real time RT-PCR. Tested ticks included females immediately after egg laying and unfed larvae or nymphs.

### 2.3. Tick Infection by Artificial Feeding

The system we used to infect ticks was previously described in [8]. Briefly, cut gerbil skin was used to feed ticks. Commercial defibrinated sheep blood was purchased from Eurobio scientific (Les Ulis, France). Before use, the blood was supplemented with 17 U/mL of heparin (Sigma-Aldrich, Saint-Louis, MO, USA), as well as gentamicin (Invitrogen, Carlsbad, CA, USA) and amphotericin B (Invitrogen, Carlsbad, CA, USA) at final concentrations of 10 mg/mL and 250 µg/mL, respectively. For infection studies, the blood was directly spiked with 10^4^ PFU/mL for KEMV. Ticks were placed in a tick chamber and enclosed with a mosquito mesh allowing them to breathe. To attract ticks, the setting was maintained at 37 °C by a circulating warmed water circuit [8]. In this study, we used 3000 *I. ricinus* larvae from France, 250 *I. ricinus* nymphs from Slovakia and 2000 *I. persulcatus* from Russia. The blood used to feed French larvae was spiked with KEMV smb1/Vero2/BSR3 and that used to feed Slovak and Russian ticks was spiked with KEMV smb1/Vero2/BSR4.

The blood was changed twice a day (morning and evening) in order to avoid any reduction in infectious virus titres in the blood as previously described [8]. Ticks were allowed to feed to repletion. Feeding experiments usually took six days, with an attachment rate to the membrane skin of around 90% and a repletion rate of 80–90% of engorged larvae [12].

### 2.4. Virus Detection Using Real-Time RT-PCR

Virus acquisition by larvae or nymphs and trans-stadial transmission from larvae to nymph or nymph to adult were assessed by testing samples of engorged larvae or nymphs (virus acquisition) and nymphs or freshly moulted adults using real time RT-PCR. The number of individuals from each of the tick species that were homogenised and tested was based on the number successfully engorged ticks obtained by artificial feeding. In order to assess virus acquisition, 30 engorged *I. ricinus* larvae from France, 4 engorged nymphs from Slovakia and 10 engorged *I. persulcatus* from Russia were used to detect the KEMV genome. The trans-stadial transmission of KEMV was assessed immediately after moulting into the next developmental stage. We therefore tested 10 *I. ricinus* from France, 4 *I. ricinus* adults from Slovakia and 5 *I. persulcatus* from Russia.

RNA was extracted from individual tick homogenates, as previously described [8], using a NucleoSpin^®^ RNA extract II kit (Macherey Nagel, Düren, Germany) according to the manufacturer’s recommendations. The genomic double-stranded RNA of KEMV was denatured by heating at 99 °C for 5 min prior to being tested using real-time RT-PCR [8]. Primers KEMV_F and KEMV_R and the probe KEMV_P [13] were used in the reaction at a final concentration of 0.5 µM each. All samples were tested in duplicate.

When viral RNA levels were deemed low in specific samples, we performed a preamplification to improve detection. Extracted RNAs were thus subjected to reverse transcription, and the resulting cDNAs were preamplified using primers KEMV_F and KEMV_R, as previously described [13]. The pre-amplified DNA was diluted in deionised water (volume to volume) and stored at −20 °C until further use. Primers KEMV_F and KEMV_R and the probe KEMV_P were then used in the PCR amplification of the target sequence, as described above.

## 3. Results and Discussion

### 3.1. KEMV Acquisition by I. ricinus and I. persulcatus Ticks and Trans-Stadial Transmission

Prior to artificial feeding, larvae and nymphs (10 *I. persulcatus* and 30 *I. ricinus,* respectively) and French female *I. ricinus* (collected immediately after egg laying) were all tested for the KEMV genome. All samples were negative.

The feeding of French larvae and Slovak nymphs lasted for 7 and 14 days, respectively. *I. persulcatus* larvae fed for up to 14 days. The total numbers of engorged ticks recovered after artificial feeding are summarised in Table 2. Our results with *I. persulcatus* ticks are in contrast with those obtained with *I. ricinus*. Indeed, engorgement rates were very low as compared with those of other ticks. It was necessary to stimulate them to attach to the gerbil skin and to feed by sprinkling faeces from actively feeding ticks. In our experience with ticks, we have observed that, when ticks successfully feed, the excreted faeces stimulate unfed ticks to attach and feed. *I. persulcatus* refusal to feed may be linked to the short span between the time we received them and the actual experimental procedure. Hence, they may not have adapted long enough to our laboratory environment, despite it being similar to that of the donor laboratory (Institute of Parasitology, Czech Academy of Sciences, České Budějovice, Czech Republic). The adult ticks that we initially received were mated and left to feed on a rabbit. The first generation of larvae that hatched in our laboratory were used in this study and resulted in fewer engorged larvae. Consequently, the number of homogenised and tested larvae and ticks of *I. persulcatus* was limited in order to keep enough ticks for the further assessment of other criteria of vector competence in particular trans-stadial transmission.

The first criterion of vector competence was assessed by testing the presence of the KEMV genome within the engorged ticks. In order to ensure that the detected KEMV RNA did not result from the blood within the intestinal lumen, engorged ticks were tested one week after feeding. Being heterophagous, blood digestion in ticks occurs within gut cells [14], which are sites of virus replication. The presence of the KEMV genome in engorged *I. ricinus* ticks (larvae and nymphs) was confirmed using real-time RT-PCR. Our initial real-time RT-PCR assay using the RNA extracts from French *I. ricinus* indicated low levels of KEMV RNA. Thus, in order to improve detection, the RNA extracts from the French ticks were subjected to reverse transcription and preamplification as described above. As shown in Table 2, 88.3% of the tested larvae from France and 100% of the tested nymphs from Slovakia were positive. KEMV was also found in 100% of the tested *I. persulcatus* larvae (Table 2). Even though fewer *I. persulcatus* ticks were tested using real-time RT-PCR, the infection rates in engorged larvae were comparable between *I. persulcatus* and *I. ricinus.*

Immediately after moulting, 10 French *I. ricinus* nymphs, 4 Slovak *I. ricinus* adults (2 males and 2 females) and 5 *I. persulcatus* nymphs were tested in order to assess KEMV trans-stadial transmission. Only 10% of the French ticks were positive, which is in contrast with the Slovak and Russian ticks, where 100% of the tested ticks were positive for the KEMV genome (Table 2). The Ct values in *I. ricinus* ticks (from France and Slovakia) were similar whatever the developmental stage. For *I. persulcatus*, the Ct values in engorged larvae were higher than those in nymphs (Figure 1). The levels of viral RNA were higher in nymphs than in engorged larvae, suggesting likely virus replication in between the time the larvae fed and their moulting into nymphs. The infection rates of the nymphs showed a strong contrast between the two tick species (100% for *I. persulcatus* as compared to 10% for *I. ricinus*).

Our experimental results validate the first two criteria of vector competence, namely, KEMV acquisition and its trans-stadial transmission upon moulting, for all the *Ixodes* ticks that we used. The infection rates in French *I. ricinus* nymphs were very low (10%) as compared to those in Slovak *I. ricinus* nymphs after moulting (40%) in our previous study [8]. The genetic determinants of ticks control vector competence and influence the ability of ticks to transmit a pathogen [15]. A high genetic diversity has been observed within ticks belonging to the same species from distinct geographical regions [16], hence affecting vector competence for a given pathogen. For instance, *Ornithodoros turicata* from Texas and Florida transmitted *Borrelia turicata* to mice with varying efficiencies, with the tick population from Florida being significantly more efficient [17]. African populations of *O. erraticus* and *O. verrucosus* ticks are established vectors for the African strains of African swine fever virus (ASFV). Studies with the Eurasian strain of ASFV suggest that *O. erraticus* and *O. verrucosus* are unlikely vectors of the ASFV strains currently circulating in Eurasia [18]. In addition to the influence of vectors’ genetic diversity, the genetic variability of arboviruses plays an important role in the determinism of vector competence [19], thus affecting the levels of infection in the arthropod vector. For example, the vector competences of *Ae. albopictus* and *Ae. aegypti* were assessed for three strains of CHIKV: the East/Central/South African (ECSA) CHIKV_0621, ECSA CHIKV_115 and Asian CHIKV. It has been shown that *Ae. albopictus* transmits ECSA CHIKV_0621 with a high efficiency, while the transmission efficiencies of ECSA CHIKV_115 and Asian CHIKV are higher with *Ae. aegypti* [20]. A comparative study of vector competence of different *Ornithodoros* species (*O. moubata*, *O. erraticus* and *O. verrucosus*) using several strains of ASFV showed variable infection rates depending on the tick–virus pairs assessed [21].

### 3.2. Persistent Infection in Ticks

Contrary to insects, such as mosquitoes, which can take several blood meals during the same developmental stage, hard ticks only feed once per developmental stage. Consequently, the extrinsic incubation period of ticks is longer than that of mosquitoes. Several months have been observed between two stages, meaning that tick-borne viruses must persistently infect tick tissues during this period. In a previous study [8], we observed the clearance of the virus several months post-moulting. In the current study, we tested the ticks four months post-moulting, using real-time RT-PCR. We therefore tested a pool of 25 French nymphs, 4 individual Slovak adult ticks (2 males and 2 females) and 2 individual *I. persulcatus* nymphs. The pool of the *I. ricinus* nymph from France was found to be negative, whereas *I. ricinus* adult ticks from Slovakia and *I. persulcatus* ticks from Russia were all positive for KEMV. We therefore conclude that, similar to our previous study [8], the virus appears to have been cleared in the French *I. Ricinus*, and, thus, French *I. ricinus* ticks appear to be poorly competent for KEMV [8]. To the best of our knowledge, KEMV is not an autochthonous virus in France. In addition, in these two studies, we used French and Slovak *I. ricinus* ticks and a Russian KEMV isolate. The prevalent serotypes of KEMV in Slovakia are the Lipovnik (LIP) and Tribeč (TRB) viruses, and both were isolated from *I. ricinus* [5]. Additional studies with these serotypes are likely to provide important insights into the influence that the geographical variants of both *I. ricinus* ticks and the virus may exert on competence. The developmental stage when the tick acquires the virus may significantly impact the ability of *I. ricinus* to transmit KEMV, as suggested by the results with Slovak *I. ricinus*. KEMV acquisition and trans-stadial transmission were estimated at 100%, contrasting with previous results [8]. Four months post-moulting, adult ticks remained positive for KEMV.

## 4. Conclusions

We assessed the vector competence of *I. ricinus* from France and Slovakia and *I. persulcatus* ticks from Russia for the Kemerovo virus. We used an artificial feeding system for the tick infection. For all ticks, we validated the first two criteria of vector competence, namely, virus acquisition and trans-stadial transmission. Our results suggest specificities between viral strains and tick species/developmental stages. As for the assessment of transmission to a vertebrate host, our preliminary findings did not identify IFNAR^(−/−)^ mice as a suitable host due to the high KEMV titre required to infect them via the subcutaneous route (data not shown). For that purpose, we will endeavour to assess other suitable vertebrate models in future studies to help study the third criterion of vector competence (transmission to a naïve host).

## Figures and Tables

**Figure 1 viruses-14-01102-f001:**
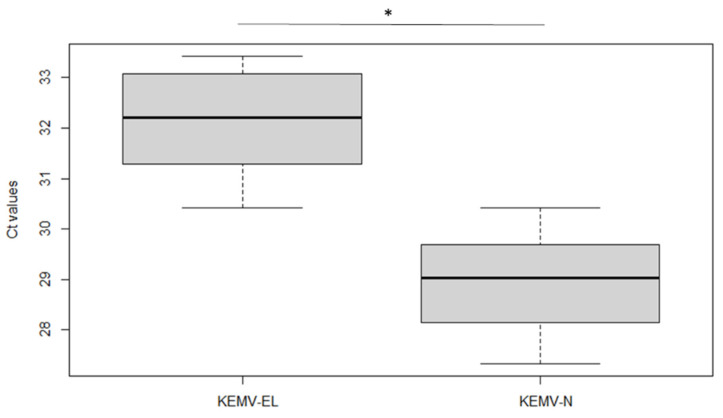
Real-time RT-PCR detection of KEMV genome segment 2 in individual *Ixodes persulcatus* tick homogenates. Ticks were fed on KEMV-spiked blood: engorged larvae (EL) and nymphs (N: resulting from moulted larvae). Mean Ct values for each group were statistically assessed using the Kruskal–Wallis test (alpha = 5%, *: *p*-value < 0.001). The lower quartile (Q1) and the upper quartile (Q3) are designated by the lower and upper lines in the graph, respectively. The darker line in the box plot designates the median.

**Table 1 viruses-14-01102-t001:** *Ixodes* ticks used for the evaluation of vector competence for KEMV.

Tick Species	Origin	Generation	Stage	Tested Stage for KEMV
*I. ricinus*	Slovakia (Institute of Zoology, Slovak Academy of Sciences, Bratislava, Slovakia)	4th generation of a laboratory colony	Nymph	Unfed nymph
	France (Provided by Sarah Bonnet, Senart forest, France)	1st generation of ticks collected in the Sénart forest from the Ile de France region	Larvae	Female after egg laying
*I. persulcatus*	Russia (Siberia) (Institute of Parasitology, Czech Academy of Sciences, České Budějovice, Czech Republic)	3rd generation of a laboratory colony	Larvae	Unfed larvae

**Table 2 viruses-14-01102-t002:** Engorgement rates of larvae and KEMV infection in engorged larvae and nymphs after artificial feeding and trans-stadial transmission.

Tested Parameters	*I. ricinus* from France	*I. ricinus* from Slovakia	*I. persulcatus* from Russia
Stage of AFS with KEMV	Larvae	Nymphs	Larvae
% of engorgement after AFS (number of engorged ticks/number of total used ticks)	22.6% (678/3000 *)	22% (55/250 *)	4.5% (90/2000 *)
% of infected ticks after AFS (number of positive ticks/number of tested engorged ticks)	88.3% (27/30 * engorged larvae)	100% (4/4 * engorged nymphs)	100% (10/10 * engorged larvae)
% of trans-stadial transmission (number of positive ticks/number of tested moulted ticks)	10% (1/10 * unfed nymphs)	100% (2/2 * unfed female 2/2 * unfed male)	100% (5/5 * unfed nymphs)

* The sample size was not determined with statistical power analysis. Rather, it was imposed by the total number of successfully engorged individuals. For instance, we used a total of about 2000 larvae of *I. persulcatus* for engorgement and only obtained 90 engorged larvae. Of the 90 engorged larvae, 10 were sacrificed and tested using real time RT-PCR. Out of the remaining 80 larvae, only 38 survived the lab conditions and moulted into nymphs. Out of these 38 nymphs, 5 were sacrificed to test the presence of KEMV genome, and the remaining 33 were kept for further assessment of trans-stadial transmission.

## Data Availability

Data is contained within the article.

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
