# Peer review of "Evaluation of Vector Competence of Ixodes Ticks for Kemerovo Virus"

_viruses, 2022, doi:10.3390/v14051102_

Round 1

Reviewer 1 Report

The revised manuscript from Migne et al evaluates the vector competence of Ixodes ticks for Kemerovo, focusing on two criteria: 1) virus acquisition, and 2) trans-stadial transmission. The removal of the mouse studies in the revised manuscript greatly improves the scientific soundness of the study while still conveying important information about vector competence. To further improve study transparency, I suggest the authors consider the following:

1. The authors' response to my original comment regarding tick sample size helps clarify why different numbers of ticks were used for each analysis. Please add the information provided in the Response to Reviewers document to Materials and Methods section 2.2 Ticks to clarify this reasoning to readers of the paper.  

Author Response

We would like to thank the reviewers for the various comments which resulted in a more coherent manuscript.

The manuscript is now clearly identified as a 'Communication' as requested. The information which we described in our previous cover letter about the sample size is now included in the manuscript as requested. We believe that the bets place to include this information is as a footnote of table 2 and we hope that this is acceptable for reviewer 1.

Reviewer 2 Report

While appreciating the commitment to the revision work conducted, the manuscript found no real improvement from these changes. many of the concerns raised by the reviewers were not adequately addressed. There are still doubts about the logic in the choice of the cohorts employed, the methodologies used and the overall organization of the manuscript. Therefore, I refer to the points raised in the previous round of review.

Author Response

We are pleased to resubmit a revised and shortened version of our communication entitled " Evaluation of vector competence of Ixodes ticks for Kemerovo virus", in agreement with your recommendations.

We would like to thank the reviewers for the various comments which resulted in a more coherent manuscript.

The manuscript is now clearly identified as a 'Communication' as requested. The information which we described in our previous cover letter about the sample size is now included in the manuscript as requested. We believe that the bets place to include this information is as a footnote of table 2 and we hope that this is acceptable for reviewers.

Concerning the detected similarities from a previous source, this is because we used paragraphs from our previously published manuscript. We apologize if this is not convenient and we have substantially modified the sections which the editorial office has highlighted as being highly similar.

As for figure 1, it does not require any copyright permission because it results from the analysis performed in the current manuscript.

Round 2

Reviewer 2 Report

Converting the manuscript to a more concise form such as a "Communication" answered many of the concerns I raised in the previous revision.
The further proposed changes further addressed the concerns raised above.

This manuscript is a resubmission of an earlier submission. The following is a list of the peer review reports and author responses from that submission.

Round 1

Reviewer 1 Report

The manuscript from Migne et al evaluates the vector competency of two different tick species, Ixodes ricinus and I. persulcatus, in supporting infection and transmission of Kemerovo virus (KEMV). The premise is important for describing vector models for future KEMV studies. However, this reviewer has reservations in the rigor of experimental design, and several revisions would strengthen the utility of the information presented in the manuscript.

Major Points:

  1. The authors evaluated different tick life stages in I. ricinus ticks from different regions for their vector competency. However, while they describe previously published findings in the Discussion, this manuscript would be greatly improved by both larval and nymph stages being evaluated for both French and Slovakian I. ricinus ticks in order to compare the relative competency of each life stage and tick origin.
  2. Similarly, why were only adult I. persulcatus ticks used? To provide a comprehensive profile, all 3 life stages of all 3 tick populations should be evaluated.
  3. The lack of a positive control in the transmission experiment reduces the ability to confidently assess the results of the Ifnar-/- mouse transmission experiments. A positive control transmission system (virus and tick population) should be provided to convince readers the lack of KEMV transmission presented in the manuscript was due to poor vector competency rather than the experimental transmission system.

Minor Points

  1. The table and graph for figure 1 are redundant, and only the table should be included.
  2. Please provide a description of how trans-stadial transmission was determined.
  3. The number of ticks and/or mice used in each experiment should be more clearly indicated in each results table. A description of the power analysis used to determine sample size should be provided in the Materials and Methods.

Reviewer 2 Report

Overview: This is a small-scale study of the vector competence of three populations (two species) of ticks for Kemorovo virus. The experimental procedures are fine but the sample sizes are quite small and the lack of transmission to mice is difficult to intepret without some measure/estimation of the amount of virus delivered during tick feeding.

Major Comments

Abstract

Second to last sentence of abstract (starting with However, the animal model….) is ambiguous- either the animal model was or was not appropriate, and this should be stated clearly.

The last sentence of the abstract is unclear, particularly given that the previous text suggests all species and stages were susceptible.  More specificity in the abstract (percentages of N infected for example) would help.

Methods

Is the titer of the BSR4 passage virus 1 x 10[9]?

Why were only the French ticks subjected to pre-amplification of DNA?

Results

Fig 1 should show data for all doses, not just those that led to detectable viremia

Sentence on line 218 contradicts preceding sentence- to what samples is the sentence on line 218 referring?

Discussion

The number of ticks assayed for trans-stadial transmission are extremely small, and thus any estimates of the efficiency of trans-stadial transmission have very wide confidence intervals and the discussion should reflect this.

The minimal infectious dose of an arbovirus is often substantially lower when delivered by a vector than an injection, and this should be noted in the Discussion.  Is there any way to estimate, even by analogy to another system, how much virus would likely have been delivered in tick saliva? 

Minor

There are a number of errors of formatting (for example the accidental ellipsis in line 37).  Also the flow and organization of the writing could be improved. For example, it is not necessary to say on line 38 that TBV are arthropod-borne viruses when their transmission has already been laid out in the previous paragraph.

Table 2: It is redundant to list number of mice per dose when the number of male and female mice is listed in subsequent columns

Section 3.1: In English, mice don’t have fingers, just toes or digits.

Throughout, use “.” rather than “,” to indicate decimals

Somewhere define acronym AFS

Reviewer 3 Report

The work concerns a topic of interest and appears to be a continuation of previous works.
In the general structure, the work has its own logic and meaning but, by the authors' own admission, there are some flaws that should be remedied to obtain a truly effective product.
If the animal model is not the real reference model (... the animal model we used was probably not optimal to confirm the third criterion of vector competence ...) it is evident that the study loses its effectiveness.
Other methodological aspects are unclear as in the case of the explanation of the different results between species with regard to engorgement rates (paragraph: lines 312-318).

The manuscript also suffers from some redundancies between introduction and discussion, which is too verbose and refers to other situations concerning "vector-virus" contexts not too appropriate to the specific case.

Minor flaws:
There are some typos and inaccuracies (the dots at line 37 or the use of the term mosquito mesh)

Reviewer 4 Report

The authors attempeted to examine the VC of different I. ricinus populations and a I. persulcatus ticks to Kemerovo virus using an artifical feeding system. Overall this study is worthwhile as it is important to characterize the ability of different tick species and populations to transmit an emerging virus with potential human and animal infections. The study design was kind of a hodge-podge of different variables; one I. ricinus population were fed as nymphs and another as larvae with a different species also being fed as larvae. Also, the different tick populations were tested by different means (i.e. some pre-amplification others without). The authors demonstrate that INFN-/- mice are susceptible to subcutaneous inoculation but do not become infected through the bite of ticks (they had previously demonstrated this). Consequently, it is difficult to interpret the data presented. Specific comments are below:

  1. It is difficult to interpret the transstadial infection rates based on the available data. Transstadial infection rates should be determined by dividing the number of positive nymphs by the number of infected larvae or number of positive adults divided by the number of positive nymphs. In this study, engorged ticks were tested immediately after detachment, therefore it is impossible to discriminate between an actual infection with replicating virus and those that are positive because of virus in the blood meal. Thus, the denominator is unknown and accurate transstadial transmission rates cannot be calculated. To accurately quantify transstadial infection rates one would have to screen previously infected flat nymphs prior to feeding on a naïve host (possibly by removing one leg) and then calculate the number of infected adults that emerge.
  2. I find it quite interesting that the I. ricinus from France do not become infected. There are numerous instances in the literature that TBVs do not have strict vector requirements (i.e. most TBVs of hard ticks can effectively establish infection of numerous other hard tick species/ genera). Consequently, I find it rather surprising that there would be VC differences between ticks of the same species. The authors do cite one study of such population differences occurring in soft ticks and another involving a bacterial pathogen.  It is possible that I ricinus is not a good vector in general. The authors show data on Ct values for KEMV infection of I persulcatus which were surprisingly low (Ct ~30), but no such data for I. ricinus. Others have shown that infection rates of POWV are significantly higher in I. scapularis (primary vector) than in D. variabilis or A. americanum.
  3. It is odd that none of the ticks transmitted to the IFNR-/- mice considering that you demonstrated that these mice are susceptible upon subcutaneous inoculation. I think it would be worthwhile to try to salivate the ticks to see if they are secreting virus in their saliva. The authors previously reported that INFR-/- mice and Balb/c were not susceptible to KEMV infection via tick inoculation; why then did they use this model for their experiments?

  4. The use and need of a pre-amplification step for the French ticks was not well justified. Further, by doing this to just the French ticks and not the others it is difficult to compare across tick species. There is no information on the limits of detection for the qPCR assay.
  5. Table 3 on line 211 should read Table 4.